# 3D-LATTE: LATENT SPACE 3D EDITING FROM TEXTUAL INSTRUCTIONS

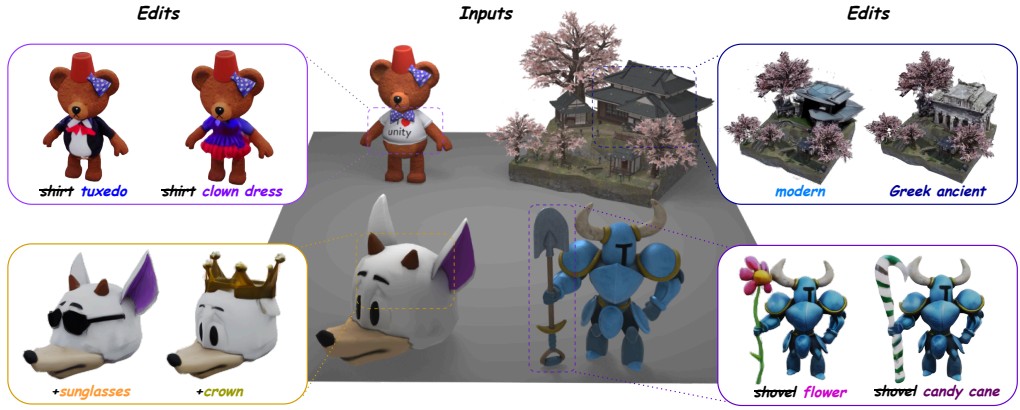

Figure 1: **3D-LATTE** takes a 3D asset and a user-specified edit instruction as input and leverages the expressiveness of the latent space of a 3D diffusion model to generate edited assets with high-quality details, geometric 3D consistency and precise adherence to diverse edit instructions.

## ABSTRACT

Despite the recent success of multi-view diffusion models for text/image-based 3D asset generation, instruction-based editing of 3D assets lacks surprisingly far behind the quality of generation models. The main reason is that recent approaches using 2D priors suffer from view-inconsistent editing signals. Going beyond 2D prior distillation methods and multi-view editing strategies, we propose a training-free editing method that operates within the latent space of a native 3D diffusion model, allowing us to directly manipulate 3D geometry. We guide the edit synthesis by blending 3D attention maps from the generation with the source object. Coupled with geometry-aware regularization guidance, a spectral modulation strategy in the Fourier domain and a refinement step for 3D enhancement, our method outperforms previous 3D editing methods enabling high-fidelity, precise, and robust edits across a wide range of shapes and semantic manipulations.

## 1 INTRODUCTION

Advances in 2D diffusion models (Rombach et al., 2021) and 3D representations, such as Neural Radiance Fields (NeRFs) (Mildenhall et al., 2020) and 3D Gaussian Splatting (3DGS) (Kerbl et al., 2023), have revolutionized 3D asset creation. More recently, powerful text-to-3D (Lin et al., 2025; Xiang et al., 2025b) and image-to-3D (Lin et al., 2025; Xiang et al., 2025b; Liu et al., 2023; 2024a) generative models have emerged, enabling scalable and expressive 3D content creation by learning from large 3D data collections (Deitke et al., 2023b;a). A core challenge in this domain is instruction-based 3D editing: modifying the geometry and appearance of a 3D object based on a language instruction, while selectively preserving the object's identity and structure. Achieving semantically accurate, multi-view consistent and high-fidelity edits has attracted considerable attention due to its importance for applications in design, virtual and augmented reality and the entertainment industry.

Many methods propose to solve this task by distilling 2D diffusion priors (Brooks et al., 2023) into a 3D representation via score distillation losses (Poole et al., 2023; Li et al., 2023; Sella et al., 2023; Koo et al., 2024) or iterative dataset updates (Haque et al., 2023; Chen et al., 2023; Chen and Wang, 2024). These approaches, however, are constrained by their reliance on 2D supervision. Specifically, they often exhibit multi-view inconsistencies, such as multi-face Janus issues, due to the limited 3D awareness, or inherit 2D editing failures from specific viewpoints, resulting in implausible 3D reconstructions. As a result, most existing methods succeed in appearance edits but struggle with large spatial or geometrical transformations that require globally consistent shape changes. A new line of work (Erkoç et al., 2024; Li et al., 2025; Huang et al., 2025) proposes to synchronously edit multiple views via multi-view diffusion priors or relies on feedforward 3D reconstruction models (Zhuang et al., 2025; Chen et al., 2024a; Hong et al., 2023) to consolidate them in 3D. Others adopt a hybrid 2D-3D approach (Chen et al., 2024b), where multi-view images are fused into a 3D representation at each denoising step. Nevertheless, existing methods based on feedforward 3D reconstruction or multi-view priors often suffer from blurry, distorted reconstructions, due to small inconsistencies across views being propagated to the 3D model. Hybrid 2D-3D methods, on the other hand, may introduce Janus artifacts and multi-view inconsistencies due to the use of 2D priors at intermediate steps.

To overcome these limitations, we take an alternative approach and leverage a 3D-native diffusion prior, where noise is injected directly to the 3D representation. This allows us to directly manipulate the appearance and geometry of a 3D asset without reliance on 2D or multi-view priors. Importantly, we operate within the model's latent space, which is structured as pixel-aligned 3D Gaussian representations (Lin et al., 2025). Motivated by the role of attention maps in 2D editing (Hertz et al., 2024), our key insight is that the 3D self- and cross-attention maps naturally capture information about the layout and composition of the 3D scene and directly model relationships between 3D Gaussians and text tokens. Building on this observation, we introduce a 3D attention injection mechanism into the denoising process of a text-conditioned 3D diffusion model. At each time step, we modulate the model's attention maps by blending or replacing them with those obtained from the generation with the source prompt, namely the description of the original, unedited asset. This allows us to guide the model to synthesize a 3D asset that semantically aligns with the edit prompt while preserving the 3D structure of the source asset.

To achieve localized edits, we leverage a vision-language model (VLM) in combination with a segmentation model to generate multi-view consistent 2D masks. These masks naturally define a 3D segmentation over the multi-view pixel-aligned 3D Gaussians, allowing us to constrain the modification to the relevant 3D regions. To enhance 3D quality, we adopt a frequency-modulated strategy that emphasizes low-frequency components early in the denoising process, encouraging the model to capture global structure before refining fine-grained details. Structural coherence is further reinforced through a geometry-aware regularization term, applied in the form of classifier guidance. Finally, to address higher-fidelity edits we adopt an iterative strategy that progressively enhances the fine-grained details of the 3D representation while preserving cross-view consistency.

To demonstrate the effectiveness of our approach, we conduct extensive user studies, report quantitative metrics based on CLIP similarity (Haque et al., 2023) and evaluate performance using GPTEval3D (Wu et al., 2024b), consistently surpassing previous state-of-the-art works.

## 2 RELATED WORKS

### 2.1 3D DIFFUSION GENERATIVE MODELS

Recent methods in 3D generation have shifted their focus toward diffusion-based models (Ho et al., 2020), which have been adapted to a plethora of 3D representations, including voxel grids (Hui et al., 2022; Xiang et al., 2025a), point clouds (Zeng et al., 2022; Nichol et al., 2022) and triplanes (Shue et al., 2023; Anciukevicius et al., 2022). Recently, methods leveraging 3DGS (Yuanbo et al., 2025; Zhou et al., 2024; Lin et al., 2025; Zhang et al., 2024) have emerged as particularly promising. GaussianCube (Zhang et al., 2024) organizes a 3DGS representation into a voxel field and applies 3D diffusion using a 3D U-Net. In contrast, DiffSplat (Lin et al., 2025) introduces a 3D latent diffusion model that directly generates 3D Gaussians by modeling an object as a set of multi-view 3DGS grids.

### 2.2 3D EDITING WITH 2D PRIORS

Editing in 3D presents unique challenges due to the need for multi-view consistency. One line of work tackles this by leveraging image-conditioned 2D diffusion models such as Instruct-

Pix2Pix (Brooks et al., 2023), injecting their editing capabilities into learned 3D representations. A pioneering work in this direction, InstructNeRF2NeRF (Haque et al., 2023), introduces the "Iterative Dataset Update" method, where rendered views from the 3D model are repeatedly updated during optimization. More recent methods, such as GaussianEditor (Chen et al., 2023) and ProEdit (Chen and Wang, 2024), build upon this paradigm. However, large edits can still lead to multi-view inconsistencies due to the lack of explicit multi-view awareness. Another line of work, such as DreamEditor (Zhuang et al., 2023), FocalDreamer (Li et al., 2023) and Vox-E (Sella et al., 2023), relies on Score Distillation Sampling (SDS) to guide 3D asset synthesis by distilling gradients from a pre-trained diffusion model. Posterior Distillation Sampling (Koo et al., 2024) improves upon SDS by aligning the latents of source and target images. While effective, these methods inherit known limitations of SDS, including oversmoothed textures, Janus artifacts, and mode-seeking behavior. As an alternative, recent methods (Wu et al., 2024a; Lee et al., 2025; Chen et al., 2024c) perform multi-view consistent edits directly on source images, which are then used to update the 3D representation. For instance, DGE (Chen et al., 2024c) leverages epipolar constraints to aggregate features across views, while GaussCTRL (Wu et al., 2024a) performs depth-conditioned 2D updates combined with cross-view alignment. However, such reliance on depth guidance can limit the model's ability to perform significant shape changes. In contrast, by operating directly within the latent space of a 3D diffusion generative model, our method enables flexible manipulation of both appearance and geometry in a fully 3D-consistent manner without relying on 2D priors or SDS-based optimization.

## 2.3 Hybrid 2D-3D Editing Methods

In parallel, another line of work has emerged that moves beyond pure 2D updates by incorporating hybrid 2D-3D strategies. SHAP-Editor (Chen et al., 2024d) learns a feed-forward editor operating in the latent space of Shap-E (Jun and Nichol, 2023) and is trained using an SDS (Poole et al., 2023) objective. However, it requires retraining for each new set of edits and relies heavily on the 2D priors and latent structure of Shap-E (Jun and Nichol, 2023). Thus, its edits lack flexibility and are often of poor visual quality, limiting its applicability. In contrast, our method generalizes across categories, produces high-fidelity outputs, and enables fast inference. MVEdit (Chen et al., 2024b) adopts a hybrid 2D-3D approach by fusing multi-view images into a 3D representation between denoising steps in a multi-view diffusion model. While this is a promising step toward enabling 3D-aware 2D edits, its reliance on 2D priors can still introduce Janus artifacts and multi-view inconsistencies.

## 3 Methodology

Given a 3D object, a source text prompt $p$ describing its original appearance and a target text prompt $p^*$ describing the desired edit, our goal is to alter the object's appearance and/or geometry so that it semantically aligns with the target prompt. At the same time, we aim to preserve regions not referenced by the prompt and maintain the original structure of the object. To this end, we introduce a zero-shot editing framework that extends attention control to the domain of 3DGS. Our method operates within the text-guided 3D diffusion model DiffSplat (Lin et al., 2025), outlined in Section 3.1. Our core idea is to inject 3D cross- and self-attention maps derived from the source 3D asset during the diffusion process that synthesizes the edited 3D asset as explained in Section 3.3. This enables semantically meaningful edits while maintaining multi-view consistency and 3D structural coherence, since our attention modifications are applied within a latent space that encodes 3D geometry. To further enhance quality, we incorporate a geometry-aware regularization mechanism (Section 3.5) and a frequency annealing strategy (Section 3.6). Finally, in Section 3.7 we introduce an iterative refinement pipeline that progressively enhances high-frequency detail and texture fidelity in the reconstructed 3D asset. An overview of our method is illustrated in Figure 2.

## 3.1 3D Representation and 3D Diffusion Model

In this work, we leverage DiffSplat (Lin et al., 2025) as our 3D diffusion backbone. In (Lin et al., 2025) a 3D object is modeled as a set of structured multi-view splat grids $\mathcal{G} = \{G_i\}_{i=1}^{V}$, where each $G_i \in \mathbb{R}^{C \times H \times W}$, $V$ is the number of input views, $H \times W$ is the spatial resolution, and $C = 12$ corresponds to the number of Gaussian attributes. Each Gaussian primitive $g_i \in \mathbb{R}^{12}$ is parameterized by its RGB color $c_i \in \mathbb{R}^3$, 3D location $x_i \in \mathbb{R}^3$, which is determined by its depth and camera parameters, scale $s_i \in \mathbb{R}^3$, rotation quaternion $r_i \in \mathbb{R}^4$, and opacity $o_i \in \mathbb{R}$. The pipeline of (Lin et al., 2025) comprises a Gaussian reconstruction module that converts multi-view RGB images, along with depth and normal maps into a Gaussian splat grid representation, a VAE that encodes the

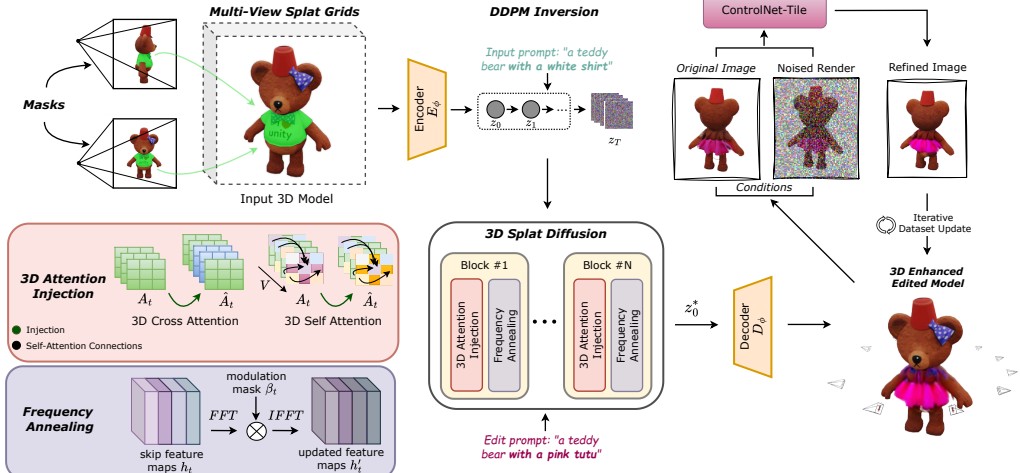

Figure 2: **Overview of 3D-LATTE**. We operate in the latent space of a pre-trained 3D generative model. The source 3D object is represented as a multi-view Gaussian splat grid and inverted into its corresponding noise latent. Starting from this latent, we perform denoising guided by the edit prompt, while injecting 3D cross- and self-attention maps derived from the source object. A geometry regularization guidance term, a frequency modulation strategy and a 3D enhancement module further refine the result. Region-specific edits are supported via masks generated using GroundingDINO (Liu et al., 2024b) and SAM2 (Ravi et al., 2024).

Gaussian splat grid into a latent space and a generative model that is trained on this representation. Finally, the reconstructed latent is fed to a VAE decoder to obtain the 3D gaussian representation, which can be rendered as multi-view images.

## 3.2 3D INVERSION

Following (Lin et al., 2025), we represent a 3D object as a set of structured multi-view splat grids $\mathcal{G} = \{G_i\}_{i=1}^V$. We use the reconstruction module from DiffSplat (Lin et al., 2025), which takes as input a set of $V$ RGB images, depth maps, and normal maps rendered from uniformly distributed camera viewpoints around the source 3D object, resulting in a 3D Gaussian for each pixel of the input views. We begin by inverting the source 3D object representation into its corresponding noise latent $z_T$. However, standard DDIM inversion (Song et al., 2021) can introduce slight errors, leading to poor reconstructions. Thus, we adapt the DDPM inversion mechanism proposed in (Huberman-Spiegelglas et al., 2024) to operate on a set of multi-view Gaussian splat grids $\mathcal{G}$ and recover a noise vector that accurately reconstructs the original 3D object.

Let the encoded Gaussian splat grids (i.e., splat latents) be denoted as $z_0 = \{z_0^i\}_{i=1}^V = \{E_\phi(G_i)\}_{i=1}^V$ where $E_\phi(\cdot)$ is the Gaussian VAE encoder of (Lin et al., 2025). We construct the forward diffusion trajectory (Song et al., 2021) and obtain the noise maps $\eta_t$ as follows:

$$z_t = \sqrt{\alpha_t}\, z_0 + \sqrt{1 - \alpha_t}\, \epsilon_t \qquad (1a) \qquad\qquad \eta_t = (z_{t-1} - \mu_\theta(z_t, t))/\sigma_t \qquad (1b)$$

where $\alpha_t$ is the cumulative diffusion noise schedule, $\epsilon_t \sim \mathcal{N}(0, I)$ is sampled independently per timestep, and $\mu_\theta$ and $\sigma_t$ are the predicted mean and standard deviation of the noise injected at each timestep, respectively. This sequence of noise maps is shown in (Huberman-Spiegelglas et al., 2024) to be edit-friendly since they are constructed with statistically independent noise samples. Starting from $z_T$, we perform the denoising process with the edit prompt $p^*$. To achieve editing while maintaining 3D structural information and layout, we apply 3D self- and cross-attention injection.

## 3.3 3D ATTENTION INJECTION

In our formulation, 3D-aware attention refers to the *self-* and *cross-attention maps* over the noisy multi-view Gaussian latents. In our 3D diffusion backbone at each timestep, features of the noisy Gaussian splat latents $\phi(z_t) \in \mathbb{R}^{V \times D \times H \times W}$ are projected into a query matrix, $Q$, while the keys, $K$, are obtained from the text prompt embeddings. Each entry $W_{i,j}$ in the 3D cross-attention map $W_{\mathcal{G}}^{\text{cross}}$ represents the influence of the $j$-th text token on the $i$-th Gaussian latent. As illustrated in Figure 3, cross-attention weights define a token-3D Gaussian splat correspondence field, enabling accurate 3D localization of interest regions. Self-attention maps $W_{\mathcal{G}}^{\text{self}}$ capture spatial and semantic relationships between all 3D Gaussian latents.

Queries and keys are computed via linear projections of the Gaussian features.

Motivated by the role of attention in encoding 3D structure and semantics within the 3D Gaussian splat representation, we introduce a 3D attention injection framework. Let $z_{t-1} = D_\theta(z_t, t, p)$ denote the denoising step with the source prompt $p$, and $z_{t-1}^* = D_\theta(z_t^*, t, p^*)$ be the denoising step with the edit prompt $p^*$. Additionally, let $W_{\mathcal{G}_t} = (W_{\mathcal{G}_t}{}^{\text{cross}}, W_{\mathcal{G}_t}{}^{\text{self}})$ and $W_{\mathcal{G}_t^*} = ((W_{\mathcal{G}_t^*})^{\text{cross}}, (W_{\mathcal{G}_t^*})^{\text{self}})$ be the attention maps computed in the denoising steps $D_\theta(z_t, t, p)$ and $D_\theta(z_t^*, t, p^*)$, respectively.

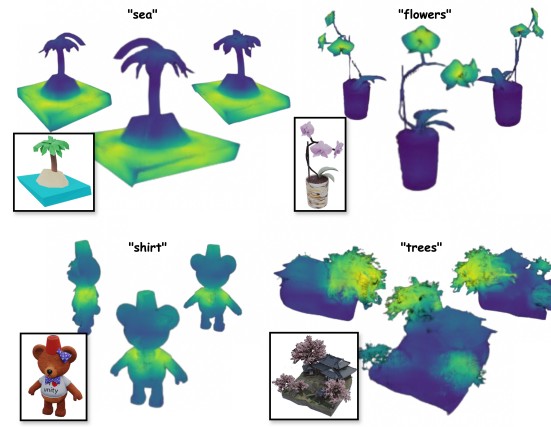

Figure 3: Per-token attention over 3D splats, rendered from multiple viewpoints.

We inject modified attention maps $\hat{W}_{\mathcal{G}_t} = (\hat{W}_{\mathcal{G}_t}{}^{\text{cross}}, \hat{W}_{\mathcal{G}_t}{}^{\text{self}})$, derived from the source prompt $p$, into the denoising process guided by the edit prompt $p^*$, as follows. At each denoising step with prompt $p^*$, we override the attention maps $W_{\mathcal{G}_t^*}$ with $\hat{W}_{\mathcal{G}_t}$ (i.e., $W_{\mathcal{G}_t^*} \leftarrow \hat{W}_{\mathcal{G}_t}$). To calculate the modified cross-attention map, we follow (Hertz et al., 2024) and inject 3D attention until a timestep $\tau_{\text{cross}}$ only on tokens shared between the original and edited prompts, using the following $F_{cross}^{3D}$ function:

$$\hat{W}_{\mathcal{G}_t}{}^{\text{cross}} = F_{cross}^{3D}(W_{\mathcal{G}}{}^{\text{cross}}, (W_{\mathcal{G}}^*)^{\text{cross}}, t)_{i,j} := \begin{cases} ((W_{\mathcal{G}_t^*})^{\text{cross}})_{i,j} & \text{if } CT(j) = None \text{ or } t < \tau_{\text{cross}} \\ (W_{\mathcal{G}_t}{}^{\text{cross}})_{i,CT(j)} & \text{otherwise} \end{cases}$$

where $CT$ is an alignment function that maps a token index from $p^*$ to the corresponding token index in $p$. In the case where there is no match, it returns $None$. By also injecting 3D self-attention during the early timesteps of the denoising process, we encourage the model to further preserve 3D structural relationships of the source asset, such as part layout, composition and spatial symmetry, before the introduction of semantic details. As illustrated in Figure 4, the spectral decomposition of the 3D self-attention graph reflects 3D scene composition and groups 3D Gaussians into distinct semantic parts. In the case of self-attention, the $F_{self}^{3D}$ function is defined as follows:

$$\hat{W}_{\mathcal{G}_t}{}^{\text{self}} = F_{self}^{3D}(W_{\mathcal{G}}{}^{\text{self}}, (W_{\mathcal{G}}^*)^{\text{self}}, t) := \begin{cases} (W_{\mathcal{G}_t^*})^{\text{self}} & \text{if } t < \tau_{\text{self}} \\ W_{\mathcal{G}_t}{}^{\text{self}} & \text{otherwise} \end{cases}$$

where $\tau_{\text{self}}$ denotes the timestep until which injection is applied.

### 3.4 MASK GENERATION AND REGION EDITING

To extract the target edit area, we formulate a query to a large vision-language model (GPT-4o) that includes the original prompt, the edit instruction, and a rendered front-view of the object. The model is asked to identify which part of the object is affected by the edit (e.g., "the shirt of the teddy bear"). We use these identified editing regions to generate 2D multi-view consistent segmentation masks. More specifically, GroundingDINO (Liu et al., 2024b)

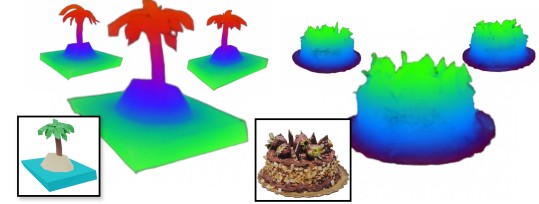

Figure 4: We form the normalized Laplacian of the 3D self-attention graph and map the first three eigenvectors to a color value per Gaussian.

processes the multi-view rendered images and the edit regions, proposed by GPT-4o to generate bounding boxes, which are subsequently refined into 2D segmentation masks tracked by SAM 2 (Ravi et al., 2024). This process is outlined in the supplementary. Benefiting from our pixel-aligned representation, these masks naturally approximate a 3D segmentation of the corresponding pixel-aligned Gaussians. To allow for flexible geometric modifications, we approximate the initial 3D mask expansion by computing the cross-attention map $\overline{W_{\mathcal{G}_t}}^{r^*}$ averaged over all time steps for the prompt token $r^*$ that refers to the desired editing region (e.g., "tutu" in Figure 2), and applying a

threshold to obtain binary masks that indicate 3D regions influenced by the new concept. The final 3D editing region $M$ is defined as the union of the original masks predicted by SAM 2 lifted to 3D and the attention-derived mask. We constrain the diffusion process to this area, by blending the original latent $z_{t-1}$ with the edited latent $z_{t-1}^*$ as follows:

$$\hat{z}_{t-1} = (1 - M) \odot z_{t-1} + M \odot z_{t-1}^* \tag{2}$$

where $\odot$ denotes element-wise multiplication. Our method also supports user-defined masks.

## 3.5 GEOMETRY REGULARIZATION

The editing signal via attention injection can introduce some uncertainty in the editing regions, resulting in Gaussians prone to semi-transparencies and premature collapse. We find that a simple geometric regularizer can help circumvent these artifacts. Thus, we incorporate a soft geometry-aware classifier guidance mechanism that stabilizes the optimization by softly penalizing the removal of Gaussians in regions that are highly relevant to the edit. To this end, we compute a soft mask $R_t^i \in [0, 1]$ for each Gaussian $i$, which reflects how relevant that Gaussian is to the current edit at denoising step $t$. We define the editing signal at each timestep as the L1 difference between the noise predictions conditioned on the edited and original prompts: $D^i = \|\epsilon_\theta(z_t, t, p^*) - \epsilon_\theta(z_t, t, p)\|_1$, where $\epsilon_\theta(z_t, t, p)$ and $\epsilon_\theta(z_t, t, p^*)$ denote the denoising model's noise predictions under the original and edited prompts, respectively. To obtain a soft mask $R_t$ that localizes the highly relevant Gaussians in each denoising step, we normalize the absolute difference values $D^i$ across all Gaussians via global min–max normalization. Thus, $R_t^i$ denotes the relevance weight of the $i$-th Gaussian, arranged in a pixel-aligned grid of size $V \times H \times W$. Our intuition is that the magnitude of this mismatch reflects the relevance of each 3D Gaussian to the edit. As the existence of Gaussians is determined by their opacity and covariance, our regularization loss is defined as:

$$\mathcal{L}_{\text{geo}} = \lambda_o \sum_i R_t^i \cdot \exp(-\gamma_o \cdot o_i) + \lambda_\Sigma \sum_i R_t^i \cdot \exp(-\gamma_\Sigma \cdot \text{Tr}(\Sigma_i)) \tag{3}$$

where $o$ and $\Sigma$ are obtained by decoding the denoised gaussian splat latents with the VAE decoder $D_\phi$ from (Lin et al., 2025). Here, $\gamma_o$ and $\gamma_\Sigma$ control the sharpness of the penalties, and $\lambda_o$, $\lambda_\Sigma$ weigh the opacity and scale terms. This formulation softly penalizes low opacity and insufficient spatial support. We incorporate our geometry-aware regularization loss as a guidance signal during denoising:

$$z_{t-1} = \hat{z}_{t-1} - s \cdot \nabla_{z_t} \mathcal{L}_{\text{geo}}(z_t) \tag{4}$$

where $s$ denotes the guidance scale, and $\hat{z}_{t-1}$ is the predicted latent from Equation 2.

## 3.6 FREQUENCY ANNEALING

The injection of attention maps from the source prompt during editing can impact the model's denoising ability, intensifying the misalignment between the condition and diffusion spaces. In some examples, this causes the model to overemphasize high-frequency visual features of the source, such as decorative textures. These may become partially preserved and degrade into artifacts on the surface of the generated 3D asset. We provide illustrations in the ablation study.
Consistent with similar observations in 2D (Si et al., 2024), in 3D generation, high-frequency components correspond to fine details while low-frequency components primarily capture 3D global geometric structure and semantic layout. Thus, we introduce a frequency annealing strategy, where we apply spectral modulation in the Fourier domain of the skip feature maps from the U-Net skip connections at each denoising step. Given the skip feature map $h_{l,t} \in \mathbb{R}^{V \times C \times H \times W}$ at layer $l$ and timestep $t$, we apply the following steps:

$$
\begin{aligned}
F(h_{l,t}) &= \text{FFT}(h_{l,t}) \\
F'(h_{l,t}) &= F(h_{l,t}) \odot \beta_{l,t} \qquad \beta_{l,t}(r) = \begin{cases} s_l, & \text{if } t > \tau \text{ and } r < r_{\text{thresh}} \\ s_h, & \text{if } t \leq \tau \text{ and } r \geq r_{\text{thresh}} \\ 1, & \text{otherwise} \end{cases} \\
h'_{l,t} &= \text{IFFT}(F'(h_{l,t}))
\end{aligned}
\tag{5}
$$

Here, $\text{FFT}(\cdot)$ and $\text{IFFT}(\cdot)$ denote the Fourier transform and its inverse. The modulation mask $\beta_{l,t}$ applies scaling based on the radius $r$. This encourages the model to boost low-frequency components during early denoising with attention injection and then transition to emphasize high-frequency components, enabling structural preservation of the source without unwanted oversmoothed textures.

## 3.7 3D ENHANCEMENT

Many existing 3D and multi-view generation models, particularly those fine-tuned on datasets with plain surface details, are constrained to low-resolution outputs and struggle to capture fine-grained geometric and appearance details. As a result, in challenging cases involving finer structures when rendering our learned 3D Gaussian representation at higher resolutions, we observe a degradation in appearance quality, characterized by blurred textures and limited details. Drawing inspiration from (Haque et al., 2023) to improve visual fidelity while preserving 3D structure and consistency, we leverage an iterative dataset update technique. Unlike prior work that primarily uses such pipelines for editing (Haque et al., 2023; Chen et al., 2023), we adapt them for 3D enhancement. Specifically, we follow an iterative process that comprises three key steps: i) rendering high-resolution views from the edited 3DGS representation, ii) enhancing these views by inputting noised version of them to the 2D diffusion backbone model, conditioned on the original edited images and, (iii) re-optimizing the 3DGS model with the updated enhanced images. To constrain the extent of correction to the editing area, the updated image is represented as: $I_{\text{blend}} = M \odot I_e + (1 - M) \odot I_{\text{src}}$, where $M$ denotes the mask containing the changed region, $I_e$ denotes the enhanced image and $I_{\text{src}}$ denotes the original image of the source object. This process is outlined in the supplementary.

As our 2D enhancement backbone, we leverage the ControlNet-Tile (Zhang et al., 2023) model, which is designed for structure-preserving super-resolution, and is effective at restoring high-frequency details and resolving visual artifacts. By progressively re-rendering and updating the images we converge to a globally consistent higher-fidelity 3D representation.

## 4 EXPERIMENTS

To evaluate our method, we construct a benchmark of 20 diverse 3D assets, each paired with three distinct edit instructions, resulting in a total of 60 samples. The assets are sourced from the Objaverse (Deitke et al., 2023b) and Google Scanned Ob-

Table 1: **Quantitative comparison using CLIP score metrics.**

| Method | CLIP Dir ↑ | CLIP Diff No-Edit ↓ |
|---|---|---|
| MVEdit (Chen et al., 2024b) | 0.11 | 0.080 |
| Vox-E (Sella et al., 2023) | 0.12 | 0.053 |
| GaussCTRL (Wu et al., 2024a) | 0.07 | **0.033** |
| PDS (Koo et al., 2024) | 0.05 | 0.097 |
| InstructGS2GS (Vachha and Haque, 2024) | 0.06 | 0.088 |
| 3D-LATTE (Ours) | **0.18** | 0.040 |

jects (GSO) (Downs et al., 2022) datasets. As main baselines, we include Vox-E (Sella et al., 2023), a voxel-based method using SDS, MVEdit (Chen et al., 2024b) and GaussCTRL (Wu et al., 2024a). For completeness, we also report quantitative results on two additional well-studied baselines: InstructGS2GS (Vachha and Haque, 2024) and Posterior Distillation Sampling (PDS) (Koo et al., 2024). For the quantitative evaluation, we adopt the CLIP directional similarity (CLIP-dir) metric (Haque et al., 2023) to quantify semantic alignment to edits by measuring direction changes in CLIP text embeddings and corresponding image embeddings of multi-view renderings. To measure how well unedited regions are preserved, we include the CLIP-diff-no-edit metric (Erkoç et al., 2024), which measures the CLIP score difference between input and output images using a modified text prompt where the edited part is replaced with a generic placeholder. Lower values indicate better shape preservation. More details are presented in the supplementary. To complement these metrics, we report results from GPTEval3D (Wu et al., 2024b), a GPT-4V-based evaluation protocol in which the model compares multi-view renderings from different methods along three criteria: Text Prompt–3D Alignment, 3D Plausibility, and Texture Details.

### 4.1 QUALITATIVE RESULTS

As illustrated in Figure 5 and 7, our method produces high-quality edits that remain faithful to the input prompt. It is capable of handling significant geometric transformations, such as turning a shovel into a flower, or a house into a Greek ancient building while preserving unedited regions and the structural integrity of the original object. As shown in Figure 5, MVEdit (Chen et al., 2024b) successfully modifies appearance but struggles with morphological changes. We also observe multi-view inconsistencies, such as two faces on the teddy bear. GaussCTRL (Wu et al., 2024a) exhibits similar limitations as it also struggles with geometric transformations, due to its reliance on depth-guidance. Vox-E (Sella et al., 2023) handles geometric changes more robustly but suffers from low quality and sometimes fails to localize edits precisely (e.g., wings on the character or shovel). Moreover, it is prone to unnatural 3D geometries due to the limited 3D awareness of SDS. Additional comparisons can be found in the supplementary.

Input                    Edit                                    Input                    Edit

A modern house ~~house~~ with pink trees

A palm tree on ~~sea~~ lava

A ~~purple~~ white couch with zebra stripes

A princess with a ~~dress~~ pink gown

A teddy bear with a ~~shirt~~ clown dress

A pot with ~~purple flowers~~ sunflowers

Figure 5: **Qualitative Results.** Our method yields high-quality 3D objects for a diverse set of edits.

## 4.2 QUANTITATIVE RESULTS

The quantitative comparison in Table 1 supports the qualitative findings that our method surpasses the baselines in terms of semantic alignment of predictions to the edited text prompts measured by the CLIP-dir score. The CLIP-no-diff score demonstrates that we preserve shapes effectively, compared to the other methods and we achieve a balanced tradeoff between editing and shape preservation. While GaussCTRL achieves a lower Diff-No-Edit score, this can be attributed to its frequent failure to apply the edits, leaving the objects unaltered. This is reflected in its significantly lower CLIP-dir score and in the qualitative examples. In Table 2, we show the results of the GPTEval3D (Wu et al., 2024b) metric in

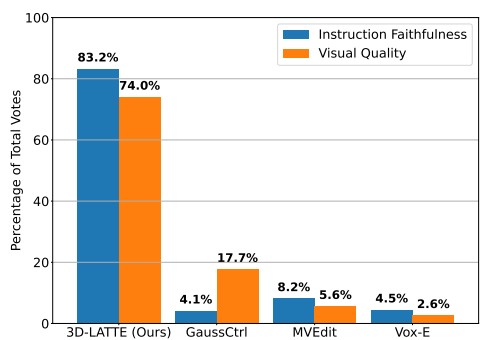

Figure 6: **User study.** Our approach shows a significantly higher percentage of votes in instruction faithfulness and visual quality.

which GPT-4V shows a distinct preference for our method. We also conduct a user study with 57 participants, who were asked to choose the best editing results among ours and the three baselines, based on two criteria: visual quality and faithfulness to the edit prompt. The results in Figure 6 show that our method was preferred by a significant margin.

Table 2: **Quantitative comparison using GPTEval3D**. Following (Erkoç et al., 2024), the scores represent the percentage of comparisons where our method wins over the respective baselines.

| Method | Prompt Algn. ↑ | 3D Plausibility ↑ | Texture ↑ |
|---|---|---|---|
| MVEdit (Chen et al., 2024b) | 89% | 72% | 71% |
| Vox-E (Sella et al., 2023) | 77% | 78% | 79% |
| GaussCTRL (Wu et al., 2024a) | 96 % | 84% | 84% |

## 4.3 ABLATION STUDY

**Effect of 3D Enhancement.** Figure 8(a) showcases the effect of our proposed 3D enhancement module. The top/bottom rows show a rendered view of the 3D objects before and after enhancement, demonstrating that the enhancement module enhances details and sharpens textures while maintaining structural coherence (e.g., sharper architectural details in the left example).

**Effect of Geometry Regularization.** We also qualitatively demonstrate the effectiveness of our geometry regularization guidance term. As shown in Figure 8(b), without this term, some edited regions become partially transparent or vanish entirely, leading to degraded geometry. The proposed regularization mitigates these issues, resulting in robust edits.

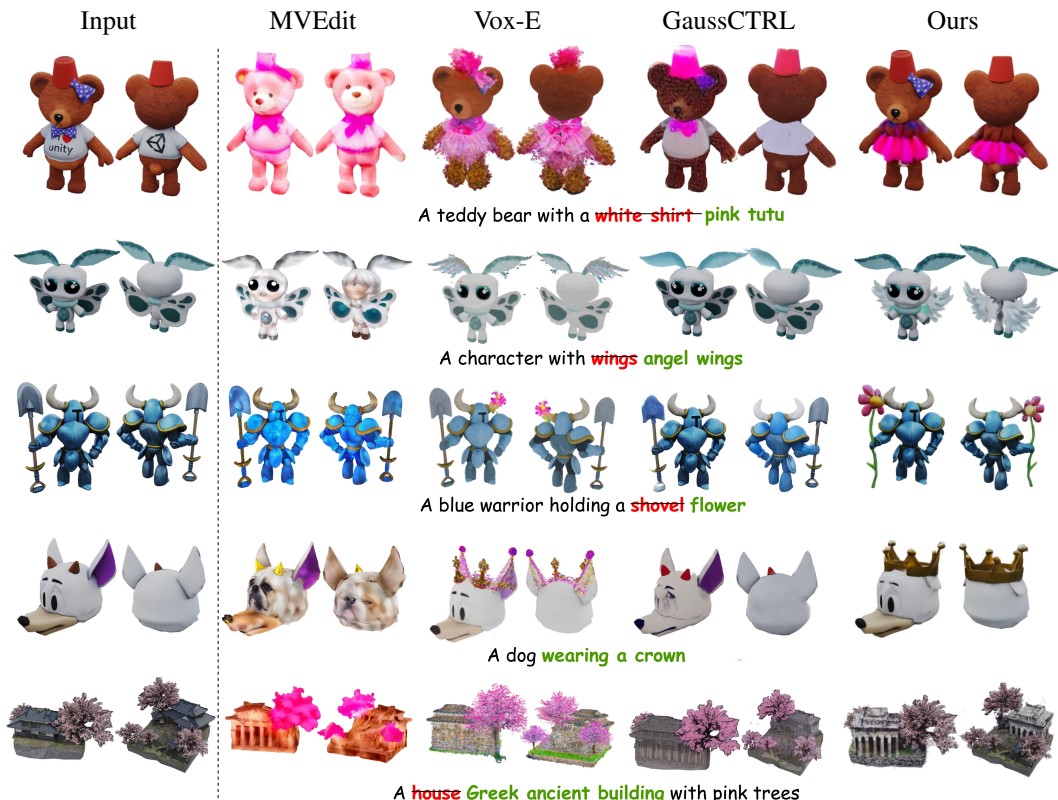

Figure 7: **Qualitative comparison with baselines.** Our approach achieves the most plausible edits wrt. the input instruction text, while preserving the unedited parts of the 3D objects.

**Effect of Frequency Annealing.** As shown in Figure 8(c), without frequency annealing, complex patterns in the source object, such as logos or prints that contain high-frequency information, can be overemphasized by the model, resulting in noisy or corrupted textures in the final edit.

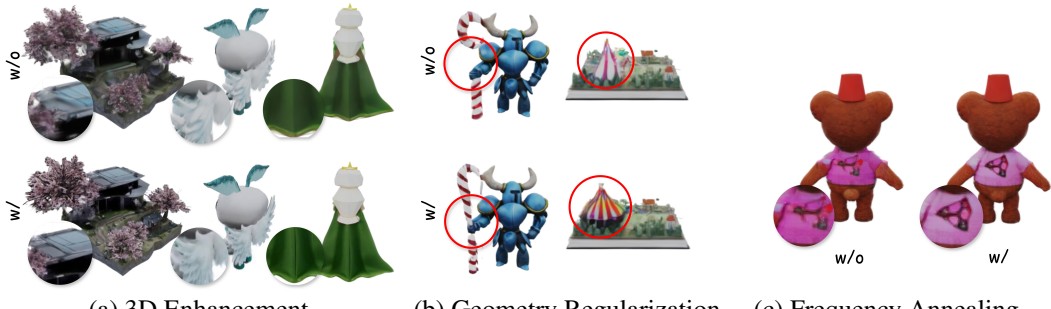

(a) 3D Enhancement.     (b) Geometry Regularization.     (c) Frequency Annealing.

Figure 8: **Qualitative ablation study.** We illustrate visual results showing the impact of the core components of our pipeline. We encourage the reader to zoom in.

## 5 CONCLUSION

In this paper, we introduced 3D-LATTE, a novel approach for instruction-based 3D editing. In contrast to existing approaches that rely on 2D distillation or multi-view edits, we leverage the expressiveness of the latent space of a native 3D diffusion model and perform editing by blending the respective 3D attention maps of the source and target object. Combined with geometry regularization, frequency annealing, and a 3D refinement step, our approach outperforms competing methods in both qualitative and quantitative evaluations and user studies.

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
