# OpenReview forum: "3D-LATTE: Latent Space 3D Editing from Textual Instructions"
_ICLR.cc/2026/Conference — ICLR 2026 Conference Withdrawn Submission_

### Official Review · Reviewer_Lp4S · 2025-10-29

**Soundness:** 2
**Presentation:** 3
**Contribution:** 2
**Rating:** 2
**Confidence:** 4

**Summary:**

The paper proposes 3D LATTE, a training free instruction-based 3D editing framework that operates entirely in the latent space of a native 3D diffusion model (DiffSplat). The core idea is to inject 3D self- and cross attention maps computed from a source prompt into the denoising trajectory guided by an edit prompt, thereby preserving global 3D structure while achieving semantic edits. The method further includes: (i) DDPM inversion adapted to multi view Gaussian-splat latents for edit friendly noise reconstruction; (ii) localized region editing via VLM-guided target-part identification and GroundingDINO + SAM2 multi view masks; (iii) a geometry aware regularizer (classifier guidance) that discourages collapse/opacity thinning in edited regions; (iv) frequency annealing (spectral modulation of skip features in Fourier space) that emphasizes low frequencies early, then high frequencies; and (v) a 3D enhancement stage that iteratively improves high frequency details using ControlNet Tile in a render–enhance–reoptimize loop restricted to the edited mask. Experiments across 20 objects (Objaverse + GSO) report superior CLIP dir / CLIP diff no edit, preferred GPTEval3D outcomes, and user-study wins versus MVEdit, Vox E, GaussCTRL, and two additional baselines.

**Strengths:**

1. The paper’s motivation is strong and clearly articulated. 3D native editing within a 3D diffusion prior (DiffSplat) avoids the canonical limitations of 2D prior distillation (SDS, iterative dataset update) and directly manipulates appearance + geometry while preserving multi view consistency.

2. Extending Prompt to Prompt–style attention control into the 3D latent space is a good  idea. The mechanism is well specified and fits the goal of preserving layout/part relationships. The spectral view of self attention as encoding 3D composition (via Laplacian eigenvectors) is insightful.

**Weaknesses:**

1. The comparisons omit several relevant recent editors (e.g., TIP Editor, SyncNoise, PrEditor3D, InterGSEdit, Robust 3D Masked editing), some of which explicitly target geometry consistency, speed/precision, or part level control, which is the problem niches this work addresses. This absence makes it harder to judge the practical advantage and may mask trade offs (speed vs. fidelity, control vs. consistency). (Paper currently compares MVEdit, Vox E, GaussCTRL, InstructGS2GS, and PDS.)

References:
* TIP-Editor: An Accurate 3D Editor Following Both Text-Prompts And Image-Prompts. SIGGRAPH, 2024.
* SyncNoise: Geometrically Consistent Noise Prediction for Text-based 3D Scene Editing, AAAI, 2025.
* PrEditor3D: Fast and Precise 3D Shape Editing, CVPR, 2025.
* InterGSEdit: Interactive 3D Gaussian Splatting Editing with 3D Geometry-Consistent Attention Prior, ICCV, 2025.
* Robust 3D-Masked Part-level Editing in 3D Gaussian Splatting with Regularized Score Distillation Sampling, ICCV, 2025.


2. While the main editing is training free, the enhancement stage performs iterative render–enhance–reoptimize using a 2D diffusion backbone (ControlNet Tile). This re introduces a 2D prior and an optimization loop with nontrivial cost; it would be useful to quantify how often it is needed and whether it ever degrades multi view consistency relative to the core method.

3. Limited dataset size and diversity. The benchmark includes 20 assets × 3 edits. It’s a good start, but relatively small for broad claims (e.g., “robust across a wide range of shapes and semantic manipulations”). More challenging categories (thin structures, glossy materials, transparent/participating media, highly concave shapes) and stress tests would increase confidence.

4. CLIP based scores and LLM based evaluations are helpful but can correlate imperfectly with true 3D consistency/geometry fidelity. There’s no analysis of multi view photometric/feature consistency, 3D geometry accuracy (when GT is available), or edit localization fidelity (IoU in 3D). The user study details (randomization, number of views presented, stats tests) are not fully described in the main text.

5. The mask pipeline uses GPT 4o + GroundingDINO + SAM2. This raises questions about replicability and failure modes when the VLM misidentifies parts or when SAM2 tracking drifts across views; it would help to document robustness and the fallback behavior. Can you report edit IoU in 3D (approximate via pixel aligned Gaussians) and failure cases?

**Questions:**

1. The token alignment part for cross attention injection and the exact timestep schedules (τ_cross, τ_self) and spectral mask parameters (r_thresh, s_l, s_h; per layer usage) should be detailedly tabulated with defaults and ablations with quantitative results.
2. For the Geometry regularizer: sensitivity to λ_o, λ_Σ, γ_o, γ_Σ, and the guidance scale s is unspecified; the risk of punishing legitimate topology/opacity changes should be analyzed.
3. Any idea how λ_o, λ_Σ, γ_o, γ_Σ, and s affect geometry fidelity and the count/opacity distribution of Gaussians in the edited region? Any cases where it over constrains legitimate removals/additions?
4. Have the authors tried attention injection on other 3D latent diffusion backbones? Are there assumptions that are specific to DiffSplat’s pixel aligned multi view grids?
5. Fig. 5 in the supp is very confusing, there is no mentioning of s_l in the main manuscript, where is this variable?

---

### Official Review · Reviewer_2HZc · 2025-10-31

**Soundness:** 3
**Presentation:** 3
**Contribution:** 3
**Rating:** 4
**Confidence:** 5

**Summary:**

This paper proposes 3D-LATTE, a method for instruction-based 3D editing that operates directly in the latent space of a 3D diffusion model (DiffSplat) rather than relying on 2D priors or score-distillation.

**Strengths:**

The authors introduce several mechanisms — 3D attention injection, geometry-aware regularization, frequency annealing, and a ControlNet-based enhancement step — to achieve high-fidelity and view-consistent edits.

**Weaknesses:**

1.Despite being labeled as “training-free,” the method involves a multi-stage, computationally heavy pipeline. It requires DDPM inversion of   3D gaussian Diffusion , Attention injection on 3D latent space, geometry-regularized update and spectural modulation, and also controlnet-tile iterative refinement using 2D diffusion. There process means significantly slower and memory-intensive editing compared to baseline methods. Although the paper proposes training-free strength, the complexity of the overall process might limit the proposed strength. To further prove the efficiency, compare the processing time and overall gpu requirement VRAM.

2. The evaluation dataset is tiny (20 objects × 3 edits), with simple text instructions (e.g., “add a crown”, “change shirt color”).
This does not convincingly demonstrate generalization across diverse 3D geometries or complex edits (e.g., structural deformation, material substitution). Comparisons are made only to 2D-prior-based baselines such as Vox-E or MVEdit, but not to other 3D-native diffusion editors (e.g., DiffGS, GaussianCube, or EditSplat). Hence, it is unclear whether the improvements arise from the 3D latent formulation or merely from using a stronger backbone. Also, there is no clue for evaluating actual edit performance such as geometric consistency using chanfer distance of volume Iou.

3. When it comes to editing 3D object, I always have some questions. Why not use reversed approach, for example, given the 3D mesh, render multiview images and apply editing on the mutlview images (or only front view), and then apply Image-to-3D model on these images.

4. What if the used segmentation model fails? What if the GPT-4o gives wrong prompt? As the method heavily relying on these third-party models, please show some failure cases or experimental evidence of using these models.

5. How to evaluate the performance gain from Frequency annealing?

**Questions:**

No

---

### Official Review · Reviewer_r6MW · 2025-11-01

**Soundness:** 3
**Presentation:** 2
**Contribution:** 3
**Rating:** 4
**Confidence:** 3

**Summary:**

For instruction-based 3D asset editing, the paper proposes a training-free framework, 3D-Latte, to address the issue of poor 3D geometric consistency. The proposed method utilizes latent-level 3D diffusion and 3D attention injection to ensure consistency between the editing instructions and model identity. Furthermore, to address quality issues such as blurring and distortion, the paper introduces geometric regularization guidance, frequency domain segmented modulation, and rendering enhancement mechanisms to achieve superior results. The experimental results surpass those of the main instruction-based 3D asset editing methods, demonstrating an advantage in CLIP metrics. The qualitative experimental results further confirm the effectiveness of our proposed method.

**Strengths:**

- The paper provides an efficient way for 3D editing by leveraging the latent space of a 3D diffusion model with  3D attention injection.

**Weaknesses:**

1. The ablations and the parameter discussion are insufficient.
- the parameters like attention duration $\tau_{cross}$ and $\tau_{self}$, frequency modulation duration $\tau$, and frequency threshold $r_{thresh}$ should be discussed.
- It would be better to show the influence of attention mechanism and add  experiments with inverse operations directly using the DiffSplat framework.

2. The paper emphasizes high fidelity and resolution in both introduction and methodology sections, but the quantitative experiments lack relevant metrics, only using related results from GPTEval3D. It would be better to include more metrics like LPIPS and  use T3Bench for comprehensive evaluation.

3. The qualitative results are still limited. For example, in Figure 5, the non-edited areas clearly show a tendency to be overly smoothed after editing. In Figure 7, the results in the first row show over-saturation; the artifacts or geometric inaccuracies appear around the "crown" in the fourth row; there is some noise in the house area in the fifth row.

4. This paper only uses 20 $\times$ 3 editing instances for evaluation, which is unconvincing. One way to increase the number of editable objects and instructions is to use Mesh2Splat and LLM.

5. It would be better to add the mask details in the main paper.

6. More comparisons are need in Tables 1 and 2. It would be better to incorporate more works like [R1] and [R2].

[R1] View-Consistent 3D Editing with Gaussian Splatting

[R2] 3D-Adapter: Geometry-Consistent Multi-View Diffusion for High-Quality 3D Generation

**Questions:**

See Weaknesses.

---

### Official Review · Reviewer_7Gn1 · 2025-11-04

**Soundness:** 3
**Presentation:** 3
**Contribution:** 2
**Rating:** 4
**Confidence:** 4

**Summary:**

The paper introduces 3D-LATTE, a training-free, instruction-based 3D editing framework that operates directly in the latent space of a native 3D diffusion model. The key mechanism is injecting 3D self/cross-attention maps of the source object into the denoising trajectory conditioned on the edit prompt, preserving geometry and multi-view consistency while applying semantic edits. The method further incorporates geometry-aware regularization, Fourier-domain frequency annealing on skip features, and an iterative 3D enhancement loop (via ControlNet-Tile) that refines rendered views and writes them back to 3DGS. Experiments on Objaverse/GSO show improved instruction faithfulness and structural preservation over prior work (e.g., MVEdit, Vox-E, GaussCTRL, PDS), supported by CLIP-based metrics, GPTEval3D, and a user study.

**Strengths:**

S1. Manipulating attention inside a 3D diffusion latent avoids many pitfalls of 2D-prior distillation (e.g., Janus artifacts, cross-view inconsistency) while keeping the pipeline training-free.

S2. The attention-injection schedule, geometry-aware guidance, and frequency annealing address common failure modes in a cohesive way.

S3. Multi-view consistent masks from VLM+GroundingDINO+SAM2 enable localized, user-intended edits; attention-based region expansion adds flexibility.

**Weaknesses:**

W1. Although presented as generally applicable, the method is in practice tightly bound to DiffSplat. Attention injection assumes its specific denoiser layout and attention block placement, the geometry-aware guidance presumes the 12-D Gaussian attribute latent on multi-view splat grids, and the frequency-annealed skip pathway is defined on splat-specific features. In effect, the paper demonstrates editing on DiffSplat rather than a backbone-agnostic principle, and the generality claim is unsubstantiated without evidence on a materially different 3D backbone (for example, NeRF-, mesh-, or point-cloud diffusion). To substantiate the method’s effectiveness and generality, the authors should evaluate it on distinct 3D backbones.

W2. The evaluation relies only on CLIP Dir and CLIP Diff No-Edit, both primarily semantic alignment measures, which does not sufficiently capture geometry fidelity, multi-view consistency, or edit locality; more importantly, the paper does not explain why the proposed method fails to reach state-of-the-art on CLIP Diff No-Edit.

W3. The ablation study relies solely on qualitative visualizations and lacks mathematical quantification and statistical significance analysis, which undermines the strength of the claims.

**Questions:**

Please see the weakness.

---

### Note · Authors · 2025-11-13

I have read and agree with the venue's withdrawal policy on behalf of myself and my co-authors.